# Antibiotic Resistance in *Klebsiella pneumoniae* and Related Enterobacterales: Molecular Mechanisms, Mobile Elements, and Therapeutic Challenges

**DOI:** 10.3390/antibiotics15010037

**Published:** 2026-01-01

**Authors:** Veronika Zdarska, Gabriele Arcari, Milan Kolar, Patrik Mlynarcik

**Affiliations:** 1Department of Microbiology, Faculty of Medicine and Dentistry, Palacky University Olomouc, Hnevotinska 3, 77515 Olomouc, Czech Republic; veronika.zdarska@upol.cz (V.Z.); milan.kolar@fnol.cz (M.K.); 2Laboratory of Medical Microbiology and Virology, University Hospital of Varese, 21100 Varese, Italy; gabriele.arcari@uninsubria.it; 3Department of Medicine and Technological Innovation, University of Insubria, 21100 Varese, Italy

**Keywords:** *Klebsiella*, Enterobacterales, antimicrobial resistance, beta-lactamases, efflux pumps, porins, two-component systems

## Abstract

Drug-resistant *Klebsiella pneumoniae* and related Enterobacterales represent an escalating global public health threat, increasingly limiting therapeutic options in both healthcare- and community-associated infections. This review summarizes how resistance in *K. pneumoniae* emerges from the synergy of intrinsic barriers and acquired determinants. Key molecular mechanisms include reduced permeability via porin remodeling (notably OmpK35/OmpK36), multidrug efflux (e.g., AcrAB-TolC and OqxAB), and enzymatic drug inactivation driven by extended-spectrum beta-lactamases and carbapenemases (e.g., KPC, OXA-48-like enzymes, and metallo-beta-lactamases). We also highlight clinically meaningful pathways underlying polymyxin/colistin resistance, including *mgrB* inactivation and PhoPQ/PmrAB-mediated lipid A modification. In addition to stable genetic resistance, adaptive programs can shape transient tolerance and persistence, including stress responses that modulate gene expression under antibiotic and host-imposed pressures. The ability of these organisms to form biofilms, particularly on medical devices, further complicates treatment and eradication. Finally, we discuss therapeutic implications and current options and limitations—including novel beta-lactam/beta-lactamase inhibitor combinations and siderophore cephalosporins—and emphasize the importance of aligning therapy and surveillance with the underlying resistance mechanisms and circulating high-risk lineages.

## 1. Introduction

The genus *Klebsiella*, classified within the family Enterobacteriaceae, includes the clinically significant species *Klebsiella pneumoniae*, a non-motile Gram-negative bacillus often surrounded by a capsule. *K. pneumoniae* was first described by Carl Friedlander in 1882 after its isolation from the lungs of a patient who had died from pneumonia [1].

*K. pneumoniae* belongs to the *K. pneumoniae* species complex (KpSC), which includes *K. pneumoniae* sensu stricto, *Klebsiella variicola*, and *Klebsiella quasipneumoniae*, among others. This taxonomic nuance matters clinically because chromosomally encoded class A beta-lactamase families differ across KpSC members (SHV [sulfhydryl reagent variable] in *K. pneumoniae*, LEN [from *K. pneumoniae* strain LEN-1] in *K. variicola*, OKP [other *K. pneumoniae* beta-lactamase] in *K. quasipneumoniae*), influencing basal beta-lactam phenotypes and the interpretation of resistance mechanisms [2].

While this review primarily focuses on antibiotic resistance in *Klebsiella*, selected examples from other clinically relevant Enterobacterales (including *Salmonella* and *Yersinia*) are discussed where they illustrate conserved resistance mechanisms, mobile genetic platforms, or clinically significant therapeutic implications.

Because antibiotic resistance in *K. pneumoniae* is tightly linked to its success as a pathogen, a brief overview of key virulence traits is provided to frame the clinical relevance of the resistance mechanisms discussed in later sections.

In particular, surface structures and hypervirulence-associated determinants that influence colonization, immune evasion, and dissemination are essential contextual factors when interpreting therapeutic failures and the spread of high-risk lineages.

The polysaccharide capsule and the lipopolysaccharide (LPS) O-antigen, identified as major virulence factors, play a crucial role in the pathogenesis of *K. pneumoniae* infections. Capsule production, associated with a lack of nutrients [3], and some capsular types’ role in immune evasion [4] further highlight their significance. O-antigens are crucial in pathogenesis, since they protect bacteria from complement-mediated killing and support *K. pneumoniae* metastatic spread to internal organs [5]. In hypervirulent *K. pneumoniae* (HvKp), additional factors (e.g., *rmpA*/*rmpA2*, aerobactin and salmochelin siderophores) underpin the propensity for metastatic spread [6].

Due to the abundance of acquired resistance mechanisms that lead to a lack of therapeutic options, *K. pneumoniae* is a major nosocomial pathogen with a global impact [7].

*Klebsiella,* represented by ‘K’ among the ESKAPE pathogens, are the primary culprits behind most healthcare-associated infections, posing a significant and immediate threat to traditional antimicrobial therapy [8,9].

*K. pneumoniae* is a leading cause of healthcare-associated infections globally. Its pathogenic potential is such that this microorganism is a relevant pathogen causing neonatal sepsis [10] and severe community-acquired infections (such as pyogenic liver abscesses, pneumonia, and meningitis) [1,11]. Moreover, *K. pneumoniae* is critical in healthcare-associated infections among elderly and immuno-compromised patients. Hospital outbreaks of *K. pneumoniae* isolates are widespread, and interhospital dissemination of resistant strains is known [12].

Notwithstanding its misleading name, *K. pneumoniae* is a leading cause of urinary tract and bloodstream infections, in addition to nosocomial pneumonia, and can also be responsible for wound and surgical site infections [1,13]. In these situations, *K. pneumoniae* is linked to differential biofilm production according to the nutritional environment [14].

Several studies demonstrate that *K. pneumoniae* develops antimicrobial resistance (AMR) through the combination of multiple mechanisms. Every *K. pneumoniae* strain carries an intrinsic *bla*_SHV_ gene, which confers resistance to ampicillin [15]. According to EUCAST expected phenotypes [16], *K. pneumoniae* is intrinsically resistant to ampicillin due to this chromosomal SHV beta-lactamase, but does not possess a chromosomal *ampC* gene. This contrasts with AmpC-risk species such as *Klebsiella aerogenes*, *Enterobacter cloacae* complex, and *Citrobacter freundii*, which harbor inducible chromosomal AmpC enzymes that significantly influence therapeutic decisions. Misclassification within the KpSC or confusion with AmpC-producing species may lead to inappropriate antibiotic selection. However, resistance to beta-lactam antibiotics beyond aminopenicillins in *K. pneumoniae* is primarily mediated by AmpC beta-lactamases, extended-spectrum beta-lactamase (ESBL) enzymes [17], and carbapenemases. A recent study identified 306 distinct beta-lactamase types across various bacterial genera, with *Klebsiella* showing the highest diversity and gene frequency for clinically relevant beta-lactamases [18]. Specifically, *Klebsiella* spp. harbor 45 unique beta-lactamase types and 11 potentially novel enzymes, underscoring their central role in the evolution and dissemination of beta-lactam resistance.

*K. pneumoniae* lacks a chromosomal *ampC* but may acquire plasmid-mediated AmpC enzymes (e.g., DHA-1 [DHA: discovered at Dhahran, Saudi Arabia]). For AmpC-risk species (e.g., *K. aerogenes*, *E. cloacae*, *C. freundii*), cefepime is suggested for invasive infections therapy; in *K. pneumoniae* with plasmid AmpC, cefepime can be considered when antimicrobial susceptibility testing (AST) supports susceptibility and optimized exposure [19]. Choice of therapy should be guided by syndrome, severity, and AST. While third-generation cephalosporins remain options for susceptible isolates in selected syndromes, invasive infections due to suspected or confirmed ESBL-producing *K. pneumoniae* are generally best treated with carbapenems; carbapenem-sparing options are reserved for specific contexts (e.g., complicated urinary tract infections with active fluoroquinolones/trimethoprim-sulfamethoxazole, or ESBL-producing Enterobacterales susceptible to cefepime/enmetazobactam in its labeled indications) [19,20]. The choice of beta-lactam therapy depends on the carbapenemase type (e.g., KPC [*K. pneumoniae* carbapenemase], OXA-48-like [OXA: active on oxacillin], or a metallo-beta-lactamase [MBL]). A detailed overview of available beta-lactam/beta-lactamase inhibitor combinations and their activity against specific carbapenemase types is provided in Table 1.

To face the growing epidemiological impact of carbapenemases in *K. pneumoniae*, starting from 2015 in the USA [21] and 2016 in the European Union [22], three new beta-lactam/beta-lactamase inhibitor combinations and one cephalosporin were approved for clinical use.

The combinations are ceftazidime/avibactam [23], meropenem/vaborbactam [24], and imipenem/relebactam [25], which share the exact bactericidal mechanism (i.e., the irreversible inhibition of the penicillin binding proteins by a beta-lactam) but differ by the structure of the inhibitor: while avibactam and relebactam are diazabicyclooctanes, vaborbactam is a cyclic boronic acid derivative. Nonetheless, these inhibitors have some intrinsic limitations: neither avibactam, nor vaborbactam, nor relebactam shows any activity against MBL (i.e., Class B carbapenemases), while vaborbactam (the only cyclic boronic acid derivative) does not inhibit Class D carbapenemases such as OXA-48 [24].

Beta-lactamases are most commonly categorized using the Ambler classification, which is based on amino-acid sequence and catalytic mechanism and divides enzymes into classes A–D. Ambler classes A, C, and D are serine beta-lactamases, whereas class B enzymes are MBLs that require one or two Zn^2+^ ions for activity [26,27]. In *Klebsiella*, clinically relevant examples include: class A (e.g., TEM [Temoneira], SHV, CTX-M-type [CTX-M: cefotaxime-hydrolyzing beta-lactamase–Munich] ESBLs, and KPC carbapenemases), class B (e.g., NDM [New Delhi metallo-beta-lactamase], VIM [Verona integron-encoded metallo-beta-lactamase], IMP [active on imipenem]), class C (e.g., plasmid-mediated AmpC such as DHA, CMY [active on cephamycins], FOX [active on cefoxitin], MOX [active on moxalactam] families), and class D (e.g., OXA-48-like carbapenemases and OXA-1-like enzymes) [18,28]. This framework is essential for interpreting resistance phenotypes and inhibitor coverage, as most currently used beta-lactamase inhibitors primarily target serine beta-lactamases, while MBLs remain a key therapeutic challenge.

Cefiderocol is a cephalosporin characterized by a catechol group that binds to ferric iron, mimicking natural siderophores that bacteria use to scavenge for iron during infection [29]. Hence, while sharing the exact mechanism of action of other beta-lactams, it can use a different route to enter the periplasmic space by adopting the so-called “Trojan horse” mechanism, bypassing resistance mechanisms like porin loss and efflux pump overactivity.

These molecules boosted the treatment options against difficult Gram-negative pathogens, including carbapenemase producers. However, in vivo development of resistance to these agents has already been documented [30] and can arise via multiple, often complementary, mechanisms. Multiple inhibitor-resistant beta-lactamase variants have been identified worldwide, especially in the KPC carbapenemase. Variations (such as amino-acid substitutions, insertions, or deletions) in two specific regions of the KPC protein, named omega- and 270-loop, affect the affinity of KPC, changing its hydrolysis patterns [31,32]. Moreover, increased copies of beta-lactamase genes on plasmids or transient amplification can lead to higher beta-lactamase enzyme levels that overwhelm inhibitors [31].

Cefiderocol resistance can occur due to alterations in the penicillin-binding protein (PBP) target, specifically PBP3 modifications [33], or, more frequently, due to depletion of siderophore transport channels supported by complementary iron acquisition systems [34].

**Table 1 antibiotics-15-00037-t001:** Overview of beta-lactam combinations and their activity against enzymes.

Combination (ATB/Inhibitor)	Active Against	Inactive/Limitations	Notes	Reference
Ceftazidime/avibactam (CZA)	KPC, OXA-48-like	MBLs	Resistance via KPC mutations (e.g., D179Y in Ω-loop), porin changes (OmpK35/36)	[23]
Meropenem/vaborbactam (MVB)	KPC-producing *Klebsiella pneumoniae*	Limited against OXA-48-like, inactive against MBLs	—	[35]
Imipenem/cilastatin/ relebactam (IMI/REL)	KPC-producing *K. pneumoniae*	Limited against OXA-48-like, inactive against MBLs	Resistance via OmpK36 disruption combined with increased expression of KPC enzymes	[36]
Cefiderocol	Broad spectrum, including MBL producers	Resistance via *cirA*/*tonB* disruption, especially in MBL backgrounds	Requires careful AST and stewardship	[29,33]
Aztreonam/avibactam (ATM/AVI)	MBL producers (also ESBL, AmpC, KPC)	—	Approved by EMA [37] and FDA [38] for cIAI, HAP/VAP, and cUTI. Resistance via PBP3 modifications (*Escherichia coli*), high-level AmpC (e.g., CMY-42), KPC mutations, porin loss, and efflux pump overexpression (*K. pneumoniae*)	[39]
Cefepime/enmetazobactam (FEP/ENM)	ESBL-producing Enterobacterales	—	Approved by EMA [40]; carbapenem-sparing option	[41]
Cefepime/taniborbactam (FEP/TAN)	KPC, OXA-48-like, some MBLs	Not approved by FDA [42]; limited clinical availability	Promising in vitro activity	[43]
Cefepime/zidebactam (FEP/ZID)	KPC, OXA-48-like, many MBL producers	—	Zidebactam binds PBP2 + has BLI activity; Phase 3 trials positive	[43]

ESBL: extended-spectrum beta-lactamase. MBL: metallo-beta-lactamase. AST: antimicrobial susceptibility testing. EMA: European Medicines Agency. FDA: U.S. Food and Drug Administration. cIAI: complicated intra-abdominal infection. cUTI: complicated urinary tract infections. HAP: hospital-acquired pneumonia. VAP: ventilator-associated pneumonia. PBP: penicillin-binding protein. BLI: beta-lactamase inhibitor.

## 2. Comprehensive Mechanisms of Antibiotic Resistance in *Klebsiella* spp.

Although many mechanisms are conserved across Enterobacterales, this section primarily focuses on *Klebsiella* spp., and other bacteria are only mentioned when they provide canonical mechanistic examples relevant to *Klebsiella*.

Antibiotic resistance in *Klebsiella* spp. is best understood as a layered phenotype, in which multiple mechanisms combine to reduce adequate drug exposure and/or neutralize the action of drugs [2,28,44]. Clinically, resistance commonly reflects (i) decreased intracellular antibiotic accumulation due to the outer membrane (OM) barrier, porin remodeling and active efflux, (ii) enzymatic drug inactivation—particularly prominent in *Klebsiella* through beta-lactamases and aminoglycoside-modifying enzymes carried on mobile genetic elements, and (iii) target-site alterations that accumulate under antimicrobial pressure and can further elevate minimum inhibitory concentration (MIC) when combined with permeability and efflux changes [28,44]. This synergy is well-documented in *K. pneumoniae*, where alterations in OmpK35/OmpK36 combined with ESBL or plasmid-mediated AmpC production can substantially increase beta-lactam MICs and contribute to treatment failure [45].

Importantly, these categories are not independent: intrinsic features (e.g., porin composition and basal efflux activity) can create a permissive background in which acquisition of ESBL/carbapenemase genes and regulatory adaptations translate into high-level resistance and therapeutic failure. For this reason, the following subsections focus on *Klebsiella*-relevant examples of target alteration, drug inactivation, reduced uptake/efflux, and stress-adaptive states (including biofilms), highlighting how these layers converge in clinical isolates [2,28].

ESBLs were first described in 1983 in *K. pneumoniae* and *Serratia marcescens* isolates in Europe [46]. Several high-impact antibiotic resistance genes (ARGs), including CTX-M, SHV-ESBL variants [47], NDM carbapenemases [48], KPC [49], and the *mcr-1*, the first mobile colistin resistance gene [50], were first characterized in clinical pathogens during the early 21st century, before achieving global distribution through mobile genetic elements.

Antimicrobial resistance in Enterobacterales, particularly in *K. pneumoniae*, represents a significant challenge for contemporary clinical care and infection control [51]. The increasing frequency of multidrug-resistant (MDR) and extensively drug-resistant (XDR) *K. pneumoniae* reflects a combination of (i) acquisition of resistance genes via successful mobile genetic platforms, (ii) dissemination of high-burden resistance plasmids, and (iii) clonal expansion of globally successful high-risk lineages [28,52].

The global burden of MDR *K. pneumoniae* is driven not only by mobile genetic elements but also by the expansion of successful epidemic lineages, often referred to as high-risk clones. High-risk clones are globally disseminated lineages with enhanced epidemic potential and long-term persistence in healthcare settings, disproportionately associated with the acquisition and stable maintenance of clinically essential resistance determinants. These lineages are disproportionately associated with healthcare transmission and the acquisition and long-term maintenance of resistance plasmids and transposons, frequently carrying ESBL and/or carbapenemase genes [2,28]. Their epidemiological success is often linked to the acquisition and persistence of epidemic resistance plasmids (frequently belonging to IncF and other clinically meaningful Inc groups), which facilitate rapid horizontal gene transfer and sustained dissemination across hospitals and regions [28,53,54].

Multilocus sequence typing commonly tracks high-risk AMR lineages and includes CG258 (notably ST258/ST512 and the related ST11). Importantly, recent genomic epidemiology highlights the growing importance of non-CG258 high-risk clones, including ST147, ST307, ST231, and ST383, which have been reported as major drivers of *bla*_KPC_ dissemination in specific European settings [55,56]. Additional globally disseminated MDR lineages repeatedly detected across regions include ST15 and ST101; ST395 has been characterized as an emerging, internationally distributed, high-risk lineage [57,58]. Notably, ST231 is also frequently linked to OXA-48/NDM producers in contemporary clinical/carriage datasets, and ST383 has shown high spreading capacity in reports describing OXA-48- and NDM-producing ST383 [59,60]. In addition, ST268 has recently garnered attention as an emerging lineage with broad geographic representation, as population genomics indicates diversification into distinct lineages and an evolutionary split between MDR- and hypervirulence-associated subclones [61].

In parallel, classical hvKp is dominated by distinct virulence-associated lineages, most commonly ST23 (often K1 capsule type) and ST65/ST86 (usually K2 capsule), which are essential to mention in reviews because the convergence of AMR and hypervirulence is increasingly documented and represents a significant clinical and public-health concern [62,63].

With this epidemiological context in mind, the following subsections move from who spreads (high-risk lineages and mobile platforms) to how resistance manifests at the mechanistic level, starting with intrinsic envelope barriers that define the baseline susceptibility of *K. pneumoniae*.

### 2.1. Mechanisms of Intrinsic Resistance

*K. pneumoniae* exhibits intrinsic resistance to ampicillin due to its chromosomally encoded SHV-1 beta-lactamase. However, developing significant resistance against broader antibiotics, such as cephalosporins and carbapenems, typically requires the acquisition of additional genetic elements such as plasmids encoding ESBLs or carbapenemases [15,48,49].

The primary mechanism of this characteristic feature is low OM permeability (Figure 1). The OM of Gram-negative bacteria acts like a selective barrier to the uptake of antibiotics. It has been compared to a molecular sieve in which the uptake of most hydrophilic molecules is size-dependent due to diffusion through the channels of water-filled porin molecules [64]. For example, hydrophobic compounds, including aminoglycosides, macrolides, rifamycins and other molecules, can infiltrate the OM via a self-promoting pathway. On the other hand, LPSs hinder the penetration of hydrophilic compounds such as beta-lactams and fluoroquinolones, making porins the primary route for these compounds to enter the cell [65]. Different types of porins in Gram-negative bacteria have been identified and categorized based on their activity (either channel or pore), their structural functionality (whether monomeric or trimeric), and their regulation and expression [66]. In addition, alterations to the OM structure, such as porin loss or changes to the phospholipid and fatty acid content of the cytoplasmic membrane, can affect the ability of a drug to penetrate the cell and contribute to the emergence of AMR [44].

LPSs, a vital component of the OM of Gram-negative bacteria, play a crucial role in maintaining membrane stability and protecting against harmful substances, including antibiotics. Comprising three sections, the O-antigen polysaccharide, the core saccharide, and lipid A, LPS is a formidable defender. In *Klebsiella*, specific genes (e.g., *mgrR*, *lpxM*, *lpxO*, *arnT*, *pagP*) targeted by PhoP and PmrA are involved in modifying LPS and reinforcing the protective function of the OM through an interaction with the LPS component of this membrane [67,68,69]. Notably, several modifications related to PhoPQ and PmrAB affect the function of the cationic antimicrobial peptide (CAP) family, which encompasses polymyxins (polymyxin B and colistin) [67].

The integrity of the cell envelope depends on a structural connection between the murein layer and a specific outer membrane protein (OMP). In Gram-negative bacteria, multiple OMPs contribute to envelope stability and permeability. OMPs are also essential for the transport of molecules and pathogenesis. The most prevalent OMPs are murein lipoprotein (Lpp), peptidoglycan-associated lipoprotein (Pal) and outer membrane protein A (OmpA). Lpp provides defense against both serum killing and neutrophil phagocytosis. A mutation in *lpp* disrupts OM permeability and increases susceptibility to anionic detergents and polypeptide antibiotics. OMPs contain a peptidoglycan-binding sequence, such as OmpA, which is necessary for the function of colicins K and L and for the stabilization of mating aggregates during conjugation. It also acts as a receptor for specific T-even-like phages and as a porin for small solutes with low permeability [70]. There is also some evidence for a role of OmpA in antimicrobial peptide resistance [71]. The Lpp and Pal, but not the OmpA, are crucial for maintaining cell envelope integrity, making the mutants susceptible to anionic detergents and polypeptide antibiotics [70]. Alterations in OMPs are considered essential for the development of clinical antibiotic resistance.

Alternative porins may become relevant when major porins (OmpK35/OmpK36) are downregulated or structurally altered (Figure 1), because they can partially compensate for nutrient uptake while also reshaping antibiotic permeability. In *K. pneumoniae*, LamB (maltoporin) and PhoE have been implicated as modulators of beta-lactam susceptibility in the context of significant porin loss [72,73]. Consistent with this, *lamB* disruption has been associated with decreased carbapenem susceptibility, whereas complementation with an unmutated *lamB* (or *ftsI*, encoding PBP3) lowered ceftazidime/avibactam MICs in induced resistant strains [72,73,74].

Three porins, OmpK36, OmpK35 and the quiescent porin OmpK37, have been described in *K. pneumoniae.* The porins OmpK35 and OmpK36, homologous to the porins OmpF and OmpC in *Escherichia coli*, play a significant role in antibiotic resistance. OmpK35 produces a significantly larger channel than its *E. coli* homolog, OmpF [75]. The functional pore of OmpK35 is slightly larger than that of OmpK36, making it easier for molecules to pass through the OmpK35 pore [76]. Previous studies have also shown that most ESBL-producing *K. pneumoniae* strains express only OmpK36, whereas most *K. pneumoniae* that do not produce ESBLs synthesize OmpK35 and OmpK36 [77]. Loss of OmpK36 is associated with cefoxitin resistance and increased resistance to oxyimino and zwitterionic cephalosporins in strains producing ESBLs and with carbapenem resistance in strains producing plasmid-mediated AmpC-type beta-lactamase. Loss of OmpK36 also results in a moderate increase in fluoroquinolone resistance in strains with altered topoisomerases (such as GyrA and ParC) and active efflux of quinolones [78,79,80,81,82].

Several alterations in the *ompK36* gene have been identified in the context of genetic mutations that significantly influence antibiotic resistance. The *ompK36* 25c > t synonymous mutation reduces translation by inducing mRNA secondary structures, thereby increasing meropenem resistance. Loop 3 insertions (e.g., GD/TD dipeptides) narrow the pore lumen, elevating carbapenem MICs to 256 mg/L. Full *ompK36* deletions yield MICs ≥ 512 mg/L but impair fitness. *ompK35* deletions under low osmolarity synergize with ESBLs, quadrupling ceftazidime MICs. Dual *ompK35/ompK36* loss amplifies carbapenem resistance (8–16× MICs) but reduces virulence [83]. Importantly, regulatory adaptations, such as *pstSCAB*-*phoU* or *phoB*-*phoR* mutations, play a crucial role in modulating PhoE expression, thereby balancing resistance and metabolic adaptation [84]. These mutations define key resistance mechanisms in clinical isolates.

Furthermore, the study of significant porins in *K. pneumoniae* revealed that inserting an IS*1380* family transposase into the region upstream of *ompK35* leads to disrupting probable promoters needed for *ompK35* expression [85]. For a comprehensive understanding of the acquired mechanisms of resistance in *K. pneumoniae*, particularly the role of OmpK35, OmpK36, NlpD, and KvrA proteins in resistance to meropenem/vaborbactam, the reader is strongly encouraged to refer to the Acquired Resistance section.

Besides the major porins, *K. pneumoniae* may express other porins, such as OmpW, OmpK26, LamB, and KpnO. KpnO is directly involved in aminoglycoside resistance. These alternative porins may be crucial for the microorganism without OmpK35/36. Their higher activity causes resistance to tobramycin, streptomycin, and spectinomycin [86]. Srinivasan et al. demonstrated the pivotal role of the CpxAR two-component signaling system in altering the expression of KpnO to mediate AMR in *K. pneumoniae*. This finding has significantly advanced our understanding of antibiotic resistance mechanisms and could potentially lead to the development of new strategies to combat resistance. The authors also showed that loss of the PhoBR-regulated porin KpnO increased AMR and reduced virulence in *K. pneumoniae* [86]. OmpK26 is a small monomeric porin that belongs to the KdgM family of porins. Insertion-duplication mutagenesis of the OmpK26 coding gene, *yjhA*, in a carbapenem-resistant, porin-deficient isolate can cause expression of OmpK36 and reversion to a carbapenem-susceptible phenotype. That makes OmpK26 indispensable for the loss of OmpK36 when bacteria become resistant to carbapenems [87]. The role of another minor porin, OmpK37, is mainly connected to carbapenem and beta-lactam resistance. However, the role of OmpK37 in antibiotic resistance is not completely clear [88,89,90].

Consistent with the layered resistance framework outlined above, reduced outer-membrane permeability and efflux contribute meaningfully to clinically relevant resistance in *K. pneumoniae* [2,28]. In particular, decreased expression or structural alteration of major porins (notably OmpK35/OmpK36), together with upregulated multidrug efflux systems, can increase MICs across multiple antibiotic classes and potentiate the impact of acquired beta-lactamases [44,45].

Reduced influx is only one side of the exposure problem; *K. pneumoniae* can further lower effective intracellular antibiotic concentrations by actively exporting diverse compounds via multidrug efflux systems. As transmembrane proteins, efflux pumps are key to bacterial survival because they transport various toxic compounds, including antibiotics, across bacterial membranes in an energy-dependent manner (Figure 1). They work synergistically with the impermeable double membrane to make these pathogens intrinsically resistant to many antibiotics. The effect of different efflux systems on specific drugs varies, with some systems conferring high levels of resistance and others low. Nonetheless, efflux is an essential “platform” mechanism that facilitates the effectiveness of most other resistance mechanisms [91]. The efflux status of cells affects the rate of evolution of resistance in a population. Lack of efflux function has been shown to decrease the frequency with which antibiotic-resistant mutants are selected [92,93,94]. Furthermore, efflux activity can lead to altered expression of other genes involved in intrinsic antibiotic resistance. For example, deletion or inhibition of *acrAB* can lead to lower expression of the OM porin OmpK35, which is homologous to OmpF in *E. coli.* As a result, membrane permeability is reduced, and intracellular drug accumulation is limited [95].

An upsurge in the expression of efflux pumps could be triggered by alterations in local repressor genes [96] or could stem from the activation of a global transcriptional regulator [97]. It has been shown that the expression of the Mex systems of *Pseudomonas aeruginosa* and the *acrAB* efflux system of *E. coli* is at its peak when the bacteria are under stress due to unfavorable conditions. However, if overexpression of efflux pumps is not controlled, it could lead to potential risks such as the loss of nutrients and metabolic intermediates [98]. These potential risks underscore the importance of understanding the mechanisms of antibiotic resistance and the urgent need for effective strategies to combat it.

Drug efflux pumps in bacteria are typically categorized into five groups: the resistance nodulation cell division (RND) family, the major facilitator superfamily (MFS), the small multidrug resistance (SMR) family, the ATP-binding cassette family, and the multidrug and toxic compound extrusion (MATE) family. Among all recognized transporters, RND pumps (such as AcrAB and OqxAB) appear to be the most prevalent in Gram-negative bacteria [99]. The MATE family is the most recent group to be classified within the MDR efflux pump families, with approximately twenty transporters identified and studied in a range of bacteria [100]. A 2015 study described the cloning of genes related to drug resistance from *K. pneumoniae* MGH785781. The authors identified a putative gene encoding a MATE-type multidrug efflux pump called *ketM***.** They described its role in active 4′,6-diamidino-2-phenyl indole efflux and demonstrated proton motive force for substrate transport by KetM [101].

AcrAB and OqxAB (Figure 1) are standard efflux pumps contributing to antibiotic resistance in *K. pneumoniae*. The *acrRAB* operon, particularly the AcrAB multidrug efflux system, plays a key role. In this operon, *acrR* encodes the AcrAB repressor. In contrast, *acrA* and *acrB* encode a periplasmic lipoprotein anchored to the inner membrane that bridges the outer and inner membranes and an integral membrane protein located in the cytoplasmic membrane [102].

In their study, Xu et al. demonstrated the presence of *oqxAB* and *acrAB* in nitrofurantoin-resistant *K. pneumoniae*. They found a single copy of the entire *rarA*-*oqxABR* locus and the *ramA*-*acrABR* locus on the chromosome of a clinical isolate. Experiments suggested that *ramA*-dependent AcrAB and OqxAB efflux pumps were implicated in nitrofurantoin resistance. The findings provide valuable insights into the mechanisms of antibiotic resistance in *K. pneumoniae* and may help in the development of new strategies to combat this resistance. For instance, observed reduction in the MIC of nitrofurantoin upon deletion of *acrB* or *oqxB* alone and further reduction in the MIC of nitrofurantoin upon simultaneous deletion of *acrB* and *oqxB* highlight the potential of targeting these efflux pumps to enhance the efficacy of nitrofurantoin [103].

Many other efflux pumps, such as KpnEF, KpnGH and KmrA, are involved in antibiotic resistance. KpnEF is an SMR-type pump that efficiently transports colistin. KpnGH is an MFS-type pump that is an intrinsic resistance determinant in *K. pneumoniae.* The ultimate section comprehensively explains how the *kpnEF*, *cpxR*, *kpnGH*, and *kmrA* genes contribute to intrinsic, acquired, and adaptive resistance.

It is worth noting that in Enterobacterales, baseline efflux activity can shape both intrinsic susceptibility and the trajectory of resistance evolution. In *K. pneumoniae*, increased expression of major multidrug efflux systems (e.g., AcrAB-TolC and OqxAB), often driven by regulatory changes affecting local repressors or global regulators, can reduce intracellular antibiotic exposure and facilitate stepwise selection of higher-level resistance when combined with target-site mutations or additional acquired determinants [2,28,44]. Beyond effects on drug accumulation, efflux activity has also been implicated in accelerating resistance acquisition by facilitating plasmid transfer in Enterobacterales (demonstrated for AcrAB-TolC in *E. coli*), providing an additional rationale to investigate efflux-targeting strategies in *K. pneumoniae* [104]. Consequently, efflux pump inhibitors (EPIs) remain an attractive adjunct concept, but their translation remains challenging due to permeability barriers, spectrum limitations, and toxicity concerns. No clinically established EPIs are currently available for the routine treatment of drug-resistant *Klebsiella* infections [44].

Additionally, efflux pump genes are remarkably preserved within a particular bacterial strain. They can also be located on mobile genetic elements such as plasmids, transposons, and integrons, thereby providing a mechanism for resistance transfer [105,106]. Because porin expression, efflux capacity, and envelope remodeling are highly plastic traits, they are best understood in conjunction with the regulatory circuits that reprogram them in response to environmental and antimicrobial stress.

A two-component system (TCS) is a common bacterial regulatory module (Figure 2) that enables adaptation to changing environments [107]. Some have been implicated in porin expression [86,108]. TCSs typically comprise a transmembrane sensor histidine kinase that receives external input signals and a cytoplasmic transcriptional response regulator that communicates a proper change in bacterial cell physiology [107]. TCS involvement in *K. pneumoniae* is frequently limited to colistin resistance [109]. Zhang et al. analyzed the correlation between TCS and OMP expression [110]. Most TCSs had a strong correlation with OMPs. The authors suggest that the TCS and OM porin in *K. pneumoniae* have a specific connection. The TCS CpxAR comprises the sensor histidine kinase CpxA and the response regulator CpxR. CpxAR regulates the expression of efflux pump- and porin-encoding genes to affect the susceptibility of *K. pneumoniae* to several antibiotics [111]. This regulation of antibiotic susceptibility by TCSs is a crucial aspect of resistance, emphasizing the urgency and importance of understanding these systems. Knocking out *cpxAR* resulted in increased antibiotic susceptibility to beta-lactams (imipenem, cefepime, ceftriaxone, ceftazidime, and cefotaxime) and chloramphenicol compared to the wild-type strain.

The TCSs PmrAB, PhoPQ, and CrrAB are critical regulators of polymyxin and colistin resistance. Deletion of both *pmrA* and *phoP* has been shown to reduce polymyxin resistance [112]. Additionally, sensor kinase *pmrB* gene mutation could upregulate *pmrA* and *pmrK* expression, affecting LPS modification (Figure 1) and resulting in colistin resistance [113]. These findings highlight the intricate regulatory roles of these TCSs in developing antibiotic resistance. For more information about TCSs, see Section 2.4. These TCS-driven programs ultimately act on OM architecture and homeostasis; therefore, the following paragraph briefly highlights additional envelope biogenesis pathways that can modulate barrier function and antibiotic susceptibility.

Beyond porin remodeling and efflux, intrinsic susceptibility in *K. pneumoniae* is also shaped by envelope biogenesis pathways that maintain OM integrity. The Lol lipoprotein trafficking system (LolCDE–LolA–LolB) and periplasmic/outer-membrane assembly factors, such as SurA and the Tol–Pal system, contribute to the correct localization and stability of envelope components; perturbation of these pathways can compromise barrier function and increase susceptibility to multiple antibiotics [114,115,116,117,118]. Under polymyxin/colistin stress, coordinated changes in envelope maintenance and trafficking pathways have been reported, supporting the view that outer-membrane barrier homeostasis is an active contributor to intrinsic resistance phenotypes [119].

Taken together, these intrinsic layers define a permissive background in which horizontally acquired resistance genes—often carried on successful plasmids and transposons—can drive rapid shifts from reduced susceptibility to clinically overt resistance in *K. pneumoniae* [2,28].

### 2.2. Mechanisms of Acquired Resistance: Horizontal Gene Transfer and Mutational Adaptations in Bacteria

The two types of acquired resistance are horizontal transfer of genetic elements and mutational resistance. DNA elements, including plasmids, transposons, integrons, prophages and resistance islands, can harbor ARGs and be acquired by conjugation, transformation, or transduction. These phenomena can increase antibiotic and multidrug resistance due to plasmids containing multiple resistance cassettes.

Horizontal gene transfer via conjugation plays a significant role in bacterial evolution, especially in spreading ARGs (Figure 1). DNA transfer is mediated by close association between donor and recipient bacteria, mainly in F-like plasmids. This process, known as mating pair stabilization (MPS), is facilitated by the interactions between the TraN protein of the OM, encoded by the plasmid in the donor and the OMPs in the recipient, which are encoded by the chromosome [120,121].

The dominant theory of conjugation in the typical F plasmid implies that a sex pilus, originating from the type IV secretion system on the donor’s surface, extends and makes contact with the recipient [122]. The pilus subsequently pulls back, bringing the recipient cell closer to the donor cell [123]. In the process of conjugation, cells form clusters called mating aggregates, which consist of two or more cells. These cells are connected by tight “mating junctions” that allow for direct contact between one cell wall and another. This is part of the process known as MPS [124]. Recently, seven TraN sequence types have been reported. They are grouped into four structural types: TraNα, TraNβ, TraNγ, and TraNδ. Moreover, there is evidence of specific pairing between TraNα and OmpW, TraNβ and OmpK36 of *K. pneumoniae*, TraNγ and OmpA, and TraNδ and OmpF. Additionally, a single amino acid insertion into loop 3 of OmpK36 influences the efficiency of TraNβ-mediated conjugation of the resistance plasmid pKpQIL in *K. pneumoniae* [121]. In *K. pneumoniae*, this conjugation-driven exchange underpins the rapid dissemination of plasmids carrying ESBL and carbapenemase genes across wards, hospitals and regions, helping to explain the speed at which high-risk lineages accumulate new resistance traits [28].

Plasmids are major vehicles of horizontal gene transfer in Enterobacterales and play a central role in the rapid acquisition and dissemination of resistance determinants in *K. pneumoniae* (Figure 1). PCR-based replicon typing distinguishes multiple plasmid types, and clinically significant resistance genes are frequently associated with a limited set of incompatibility (Inc) groups, including IncF, IncI, IncA/C, IncL (formerly IncL/M), IncN, and IncH [125].

Most new plasmid types have been shown to be associated with the predominant plasmid family known as IncF. These IncF plasmids, found in all Enterobacteriaceae species, are frequently linked to genes responsible for virulence. They are typically between 45 and 200 kb in size, have a low copy number, and contain replicons associated with FII. The standard FII replicon, made up of the *repA* gene, is a defining feature of these plasmids. Transcription of this gene is regulated by antisense RNAs that function in trans by matching sequences on the target sense mRNAs. The DNA sequences of FII replicons in plasmids from various species may differ. Still, they could all originate from a single ancestral replicon and share more than 75% similarity in their nucleotide sequences. FII replicons define the diverse IncFII plasmid family. They can be further classified into specific groups, such as FII (*E. coli*), FIIs (*Salmonella*), FIIy (*Yersinia*), and FIIk (*Klebsiella*), based on PCR and sequencing [126]. With the major genetic vehicles established, we next summarize the key resistance determinants they mobilize and the phenotypes they confer in clinical *K. pneumoniae*.

Horizontal gene transfer is crucial for the dissemination of antibiotic resistance among bacterial populations. This process mainly affects aminoglycoside and beta-lactam resistance in *K. pneumoniae* but has also been observed in several other classes. Aminoglycoside-modifying enzymes, proteins produced by bacteria that can inactivate aminoglycosides by chemically modifying them, are located on mobile genetic elements. This leads to a reduced affinity for the 30S ribosomal subunit, the main target of aminoglycosides [127]. Glycosyltransferases and phosphotransferases may be the main causes of rifamycin resistance. If these enzymes are expressed, they could potentially decrease the organism’s vulnerability, especially if the phosphotransferases are located on mobile genetic components such as plasmids, where their expression levels can be quite high [128]. An excellent example of horizontally transferred resistance is the *qnrA* and *qnrB* genes from the marine bacterium *Shewanella,* which have also been identified in *K. pneumoniae*. These plasmid genes confer resistance to quinolones [129,130].

Efflux pumps, the unsung heroes of antibiotic resistance, play a significant role in exporting antibiotics, other substances, such as dyes and detergents, and host-derived antimicrobial agents [131]. Their inhibition has been proven to have a detrimental effect on bacterial pathogenesis, a process by which bacteria cause disease in a host, underscoring the urgency of understanding and addressing the role of efflux pumps in antibiotic resistance [132,133,134,135].

Essential resistance genes located in the core genome of *K. pneumoniae* include *fosA* and the efflux pump *oqxAB*, which exhibit low resistance to fosfomycin and the quinolone nalidixic acid. These resistance genes are transferred horizontally, mainly carried by plasmids [64]. The acquisition of an MDR plasmid in *K. pneumoniae*, which carries multiple genes conferring resistance to different antibiotics, has been shown to cause increased transcription of efflux genes, further strengthening the link between MDR plasmids and efflux [136]. However, induced expression of intrinsic efflux pumps is also one of the most common forms of mutational resistance. Generally, spontaneous mutation frequencies vary among antibiotics, with resistance frequencies ranging from 10^−6^ to 10^−9^ for individual antibiotics. The mutation rate can further increase under certain conditions, such as in the presence of DNA-damaging agents [137].

The pressing issue of carbapenem resistance, a rapidly escalating problem primarily driven by the accessory genome or combined with core genome mutations, demands immediate attention. Carbapenem resistance in *K. pneumoniae* can be mediated through the upregulation of efflux pumps [138], alteration of OM porins in the core genome [73], and hyperproduction of ESBL enzymes or AmpC beta-lactamases in the accessory genome [139]. The dissemination of the carbapenemase genes *bla*_KPC_, *bla*_NDM-1_, *bla*_OXA-48_ and the ESBL gene *bla*_CTX-M-15_ is a significant concern. These genes are associated with transposons embedded in plasmids that can spread to other strains, genera, species, and even the bacterial chromosome [140,141,142].

As mentioned above, induced expression of intrinsic efflux pumps, such as those encoded by *acrAB* and *oqxAB*, has also been associated with reduced susceptibility to tigecycline, fluoroquinolones, and other antimicrobials [102,143]. Specifically, the *oqxAB* gene is generally located on chromosomes and plasmids and confers low to intermediate resistance to quinoxalines, quinolones, tigecycline, and nitrofurantoin. It is highly probable that the *oqxAB*, carried on a plasmid, was obtained from the chromosome of *K. pneumoniae* [144]. Several plasmid replicons, such as IncF, IncH, IncI, IncHI2, and IncX, could transmit the *oqxAB* gene [145,146,147]. It can propagate concurrently with other ARGs, such as *bla*_CTX-M_, *rmtB*, and *aac(6′)-Ib*, as well as virulence genes and genes resistant to heavy metals. In the regulation of OqxAB expression, both RarA, functioning as an activator, and OqxR, acting as a repressor, are known to play pivotal roles [148].

Several studies have reported a correlation between reduced susceptibility to quinolones and AcrA overexpression in several quinolone-resistant clinical *K. pneumoniae* strains. However, the genetic basis of this overexpression has not been described [149,150]. Furthermore, the expression of the *acrAB* multidrug exporter system is induced by various treatments with fatty acids, sodium chloride, or ethanol [151]. Additionally, it has been shown that increased AcrAB efflux pump expression in fluoroquinolone-resistant *K. pneumoniae* strains is caused by either mutation in the AcrAB repressor, AcrR, or overexpression of the transcriptional regulator RamA [152]. In addition, a study conducted by another research group on *E. coli* strains found that *E. coli* mutants lacking efflux pump components developed resistance to CIP due to specific *gyrA* and *parC* mutations. However, the resistance was low-level unless compensated by a significant increase in the expression of the alternative efflux pump genes *acrE* and *acrF*. This highlights the crucial role of an intact AcrAB-TolC efflux pump in the development of bacterial resistance [153].

The AcrAB efflux system plays a key role in antibiotic resistance by actively exporting antibiotics from the bacterial cell, lowering their intracellular concentration and diminishing their efficacy. Additionally, the *ramA* gene contributes to resistance by downregulating porin expression. Notably, no mutations have been identified within the *ramA* gene that would account for its increased expression [154]. However, further studies have demonstrated that mutations in the *ramR* gene—which normally acts to suppress *ramA* expression—lead to an upregulation of *ramA* [155].

A recent study has provided two key insights into meropenem/vaborbactam resistance in *K. pneumoniae*. The first significant discovery is the role of loss-of-function mutations in the OmpK35 and OmpK36 porins, which have been identified as key contributors to this resistance. The second finding is the impact of the loss of NlpD and KvrA proteins, particularly KvrA, on the production of the OmpK35 and OmpK36 porins. This loss reduces the susceptibility of *K. pneumoniae* isolates to meropenem/vaborbactam, further emphasizing the importance of these proteins in the context of antibiotic resistance [156]. KvrA, a transcriptional repressor, plays a critical role in *K. pneumoniae* capsulation [157], while NlpD is involved in peptidoglycan remodeling during cell division [158].

The prevalence of tetracycline resistance, mainly due to tetracycline-inactivating enzymes that catalyze the oxidation of tetracyclines, is a cause for concern. The Tet(X) family, a well-known culprit, is found in many different classes of bacteria. It can move horizontally on transposable elements, contributing to high levels of tetracycline resistance. The Tet(X) family is a group of enzymes that specifically inactivate tetracycline antibiotics, rendering them ineffective against the bacteria. The widespread distribution of *tet(X)* genes, commonly found in MDR bacteria from various environments, is a stark reminder of the magnitude of the problem, especially in light of tetracycline use. This underscores the gravity of the situation and the need for our continued research and efforts in this area [159,160,161].

Another article describes the easy dissemination of lincosamide resistance genes (e.g., *cfr*, *ermB*, *lnuA*, *lnuB*, *lsaB*, *salA*, *vgaA*) across different bacterial strains and species due to their location on mobile genetic elements, as well as the facilitation of their persistence under selective pressure from other antimicrobial agents [162]. Selective pressure refers to the conditions that favor the survival of organisms with specific traits, such as resistance to antimicrobial agents. It is a key factor in the evolution of antibiotic resistance.

KPC is associated with a wide range of plasmids that are characterized by Tn4401, a 10 kbp Tn3-like transposon with five isoforms [50,163]. OXA carbapenemases are class D enzymes characterized by their ability to hydrolyze cloxacillin or oxacillin. The plasmid-encoded OXA-48 is found in *K. pneumoniae* and confers a high level of resistance to imipenem; Tn-1999 characterizes it and is most commonly associated with IncL/M plasmids [164,165]. NDM-1 is part of many plasmids [48]. They belong to class B MBL characterized by a requirement for zinc at their active site. Infections caused by MBL-producing strains are frequently associated with travel and hospitalization in endemic regions [166]. VIM and IMP also occur in class B. Besides aztreonam, these enzymes can hydrolyze penicillins, cephalosporins, monobactams, and carbapenems. Both genes are carried on integrons and can be encoded on either chromosome or are plasmid-mediated [167,168]. ISEcp1 characterizes CTX-M-15 and is mainly associated with IncFII-FIA and IncH plasmids, which could carry other resistance genes [169,170].

If bacteria are fluoroquinolone- and carbapenem-resistant, the treatment options are severely limited, with tigecycline or colistin being the only viable choices [171]. Colistin resistance in *K. pneumoniae* is commonly caused by mutations in the core genome [172], typically through mutations in regulatory genes such as *mgrB* (Figure 1). This gene regulates the modification of bacterial lipid A, the target of polymyxin antibiotics, and decreases polymyxin activity [163]. Insertions, deletions, or nonsense mutations can inactivate the *mgrB* gene, leading to upregulation of the PhoQ/PhoP system and *pmrHFIJKLM* operon [137,173,174]. Another mechanism of resistance is modification of the chromosomal *crrB* gene and acquisition of the plasmid-borne genes *mcr-1* or *mcr-1.2* [50,175]. Since the first report of *mcr-1*, eleven additional *mcr* homologues (*mcr-2* to *mcr-12*; https://www.ncbi.nlm.nih.gov/pathogens/refgene/#mcr; accessed on 1 December 2025) encoded on different plasmids and widely distributed in Enterobacteriaceae have been reported globally, underscoring the global nature of this issue [176].

Significant developments in genomic methods, especially next-generation sequencing, have allowed us to define mutants to be generated in species previously tractable for manipulation. High-density transposon mutant libraries have been used to screen whole genomes for changes related to antibiotic susceptibility at base-pair resolution. Many genes with minor contributions to resistance have been discovered with these methods [177]. These genes, which form the “secondary resistome”, are not overlooked. Their significance in bacterial resistance mechanisms is profound and often underappreciated, making them a crucial and significant aspect of our understanding of antibiotic resistance. A study of genes involved in colistin susceptibility in *K. pneumoniae* showed that inactivation of the non-essential gene *dedA* reversed AMR in isolates with high MICs [177].

The genome of *K. pneumoniae*, a complex entity, is on average 5.5 Mbp in size and encodes 5500 genes. The core genome encodes less than 2000 genes and is part of this intricate picture. The remaining 3500 genes, known as ‘accessory’ genes, are derived from more than 30,000 protein-coding genes, highlighting the profound complexity of this organism [178]. This is a significant challenge that underscores the need for further research as we strive to understand the full extent of *K. pneumoniae*’s genetic makeup and its implications for antibiotic resistance. The intricate nature of the *K. pneumoniae* genome poses a significant challenge in our quest to fully comprehend its genetic makeup and its role in antibiotic resistance.

In the bacterial genome of *K. pneumoniae* (deposited under accession number SAMN06040388; sequence length 6.37 Mb) with complete assembly levels, we found a total of 6714 genes (5533 genes on the chromosome and the remaining 1181 genes encoded by plasmids). The data were retrieved from the NCBI Genome database (https://www.ncbi.nlm.nih.gov/datasets/genome/GCF_002903025.1/; accessed on 6 January 2025).

Analyses of publicly available genomic data from databases, such as RefSeq (accessed 5 January 2025), indicate a substantial diversity of acquired ARGs within *K. pneumoniae* (Table 2). These ARGs span multiple antibiotic classes, including aminoglycosides, beta-lactams, colistin, fluoroquinolones, fosfomycin, macrolide-lincosamide-streptogramin, phenicols, rifamycins, sulfonamides, trimethoprim, tetracyclines, and tigecycline. This diversity reflects the well-established capacity of *Klebsiella* species to harbor multiple plasmids carrying distinct sets of resistance determinants, which contribute to broad-spectrum AMR [125,179]. Additionally, other beta-lactamase genes, such as *bla*_IMI-2_ (GenPeptID: AQZ36626.1) and *bla*_TLA-like_ (GenPeptID: WP_129897325), have been reported in species outside *K. pneumoniae*, including *K. variicola* and *Klebsiella michiganensis*, respectively. For readability, Table 2 collates the acquired resistance genes most frequently reported in *K. pneumoniae* and links them to the corresponding target drug classes.

Significantly, genotype alone does not entirely predict therapeutic response, as exposure to antibiotics and host-imposed stresses can trigger reversible transcriptional and physiological programs that transiently alter susceptibility, tolerance, and persistence, even without the acquisition of new genes [44,180].

### 2.3. Mechanisms of Adaptive Resistance

It is widely acknowledged that bacteria encounter various changing conditions and stresses in their natural habitats. Their adaptive mechanisms trigger specific, highly regulated protective responses affecting innate antimicrobial susceptibility. For instance, exposure to reactive oxygen and nitrogen species (oxidative/nitrosative stress), membrane damage (envelope stress) and DNA damage all impact bacterial susceptibility to various antimicrobial drugs by initiating stress responses that positively influence the recruitment of resistance determinants or promote physiological changes that compromise antimicrobial activity [181]. In this context, bacterial stress responses—such as the activation of RND-family multidrug efflux systems and the induction of the stringent response mediated by elevated levels of the alarmone (p)ppGpp (guanosine tetra—and pentaphosphate)—have been identified as additional determinants contributing to reduced antibiotic susceptibility [181,182]. The (p)ppGpp alarmone, produced by RelA/SpoT-family enzymes in response to nutritional limitation and other stresses, can remodel global transcriptional programs, slow growth, and promote transient tolerance phenotypes that may potentiate classical resistance mechanisms. These responses shield bacteria from stress and induce changes that affect their antibiotic susceptibility. This knowledge is crucial for researchers, microbiologists, and healthcare professionals interested in antibiotic resistance as it enhances their understanding of the complex interplay between bacteria and antibiotics. One of the most recognized mechanisms involved is the two-component regulatory system [181].

Adaptation, the progressive modifications bacteria undergo to enhance their tolerance in stressful environments, is a critical aspect of their survival strategy [180,183]. In *K. pneumoniae*, these adaptive mechanisms improve bacterial fitness and pathogenic potential by altering the infection strategy [184] and conferring a transient resistance to the stressors. These mechanisms, often referred to as “adaptive resistance”, are a form of adaptation that results in transient resistance during drug exposure due to alterations in the expression of genes and proteins [185]. It should be noted that because adaptive resistance is transient, it typically reverts when the inducing condition is removed. This contrasts with the characteristics of intrinsic and acquired resistance mechanisms, which are stable and transmissible from generation to generation [186] and need to be distinguished from resistance in heterogeneous subpopulations [187].

It has been firmly established that methylation is crucial in foreign DNA restriction and gene expression [188]. Changes in DNA methylation patterns, often observed when bacteria are exposed to antimicrobials, can lead to transient adaptive antibiotic resistance, including strains that overproduce efflux pumps. Accordingly, methylation-mediated regulatory shifts should be considered among the contributors to transient adaptive resistance phenotypes observed under antimicrobial pressure [188].

The implications of adaptive resistance in *K. pneumoniae* are significant, as it can reduce susceptibility to several last-resort drugs in otherwise susceptible strains. Specifically, in carbapenemase-producing *K. pneumoniae* strains, adaptive resistance mechanisms can impair the efficacy of colistin when exposure to this drug is low [189]; a similar effect can be observed with tigecycline. When this last-resort tetracycline antibiotic is used alone, the expression levels of *acrB* (an efflux pump gene) and *ramA* (its regulatory gene) [190] are significantly higher [191]. Interestingly, although bacteriostatic–bactericidal antagonism has often been reported [192], adaptive hyperexpression of these genes could contribute to the paradoxical observation that tigecycline-resistant isolates display increased susceptibility to aminoglycosides, and that combining tigecycline with an aminoglycoside may enhance activity [191].

The role of adaptive resistance in KPC-producing strains is of paramount importance, particularly in the context of carbapenem resistance. The *bla*_KPC_ gene, a key factor in carbapenem resistance, is a necessary but insufficient condition for high MICs for carbapenems [193]. According to this hypothesis, the presence of KPC allows the hydrolysis of a relatively small number of carbapenem molecules entering the periplasmic space, thereby enabling only some subpopulations to survive. Continued entry of these beta-lactams causes peptidoglycan damage and consequent stress, which may facilitate the adaptive response (e.g., limiting exposure to OM porins) [194].

*K. pneumoniae* displays remarkable adaptability, necessitating the development of adaptive mechanisms for stable acquisition of MDR plasmids [195]. The canonical *bla*_KPC_ gene carrying pKpQIL IncF plasmid serves as a prime example of how, after plasmid acquisition, the occurrence of intergenic mutations in the genome affecting transcription and translation can be advantageous for maintaining the plasmid without loss of fitness [136]. The transcription regulator KbvR in *K. pneumoniae* is a critical factor in its virulence, defense against the immune system, and antibiotic resistance. Deletion of KbvR reduces the bacterium’s resistance by decreasing the production of capsular polysaccharide (CPS) and some OMPs [196]. The ability of the bacterium to sense and adapt to environmental osmotic stress can alter antibiotic resistance. Under high osmotic stress, KbvR is upregulated and directly regulates the expression of OmpK36. This stress is sensed by EnvZ, the sensor kinase of the EnvZ/OmpR two-component signal transduction system, which increases the activity of the OmpR regulator and the expression of OmpK36 [197]. This adaptation to high osmotic stress alters the susceptibility of *K. pneumoniae* to antibiotics, offering potential strategies for sensitizing bacteria to antibiotics and adapting them to diverse environmental challenges, providing a hopeful outlook for future solutions in the fight against antibiotic resistance.

Adaptive resistance extends beyond antibiotics and limits the efficacy of disinfectant molecules with bactericidal properties, such as chlorhexidine. The potential misuse of this compound, widely employed to prevent the spread of bacteria in nosocomial environments, may exert selective pressure on bacteria, accelerating the emergence of resistance. Reduced susceptibility to chlorhexidine in Gram-negative bacteria has been linked to multidrug efflux pumps such as CepA and QacE, and hyperexpression of *cepA* has been identified as a mechanism of resistance [175]. Besides antimicrobial compounds, adaptive resistance can be elicited by subinhibitory concentrations of molecules that do not have direct antimicrobial properties. Diazepam, a benzodiazepine drug, has been shown to reduce porin expression and induce efflux systems, thereby decreasing susceptibility to beta-lactams and other antibacterials [194].

Because these adaptive states often co-occur with intrinsic permeability changes and acquired resistance genes in clinical isolates, the resulting phenotype is best understood as an interaction of layers rather than isolated mechanisms, as outlined below [2,28].

### 2.4. Interplay of Intrinsic, Acquired, and Adaptive Resistance Mechanisms in Bacterial Survival

The intricate interplay between intrinsic, acquired, and adaptive resistance mechanisms of bacteria is exemplified by various regulatory systems and genetic adaptations that enhance bacterial survival and antibiotic resistance. One critical aspect of this interplay is the regulation mechanisms of TCSs. The CpxAR system, a key player that responds to stressors affecting protein folding in the cell envelope and thereby contributing to envelope maintenance, plays a pivotal role in bacterial survival and antibiotic resistance. Its significance lies in its ability to regulate the expression of efflux genes like *acrB*, *acrD* and *eefB*, which are responsible for pumping antibiotics out of the bacterial cell. This system is a critical component in the complex network of bacterial defense mechanisms, and understanding its function is crucial in the fight against antibiotic resistance. For instance, in *E. coli*, activation of the CpxAR-dependent response decreases susceptibility to beta-lactams, aminoglycosides, novobiocin, and CAPs. Deletion of *cpxAR* results in lower expression levels of efflux genes such as *acrB*, *acrD* and *eefB* in the mutant compared to the wild type [111].

Disruption of the *kpnEF* efflux system has been associated with increased susceptibility to multiple antibiotic classes (including cefepime, ceftriaxone, colistin, tetracycline, and streptomycin), underscoring how efflux regulation can shape resistance phenotypes beyond acquired enzymes [198]. Experimental evidence further suggests that CpxR binds to the *kpnEF* promoter, linking envelope stress sensing with efflux expression [198].

When deleted, the TCS regulator CpxR is critical for increasing bacterial susceptibility to carbapenems. CpxR binds directly to the *bla*_KPC_ promoter DNA, upregulating its transcription and facilitating the transfer of the *bla*_KPC_-carrying plasmid between *K. pneumoniae* and *E. coli* isolates. This highlights the crucial role of CpxR in regulating carbapenem resistance in *K. pneumoniae* [199]. Further, insertional inactivation of *kpnGH* results in increased susceptibility to antibiotics such as azithromycin, ceftazidime, CIP, ertapenem, erythromycin, gentamicin, imipenem, ticarcillin, norfloxacin, polymyxin B, piperacillin, spectinomycin, tobramycin, and streptomycin [186]. KmrA is another member of the MFS efflux pump family. Previous research by Li and Ge found that upregulation of the efflux pump KmrA conferred enhanced antibiotic resistance. Interrelated amino-acid substitutions (simultaneously occurring substitutions) in the *kmrA* gene can increase drug export in bacteria. The authors identified two variants with triple amino-acid substitutions near the periplasmic side and confirmed their roles in enhancing multidrug resistance [200].

PhoBR TCSs comprise the sensor histidine kinase PhoR and the response regulator PhoB (acting as a positive regulator for the phosphate regulon, becoming active when phosphate is scarce) and regulate antibiotic resistance in *K. pneumoniae*. The PhoBR TCS in *Klebsiella* facilitated resistance to gastrointestinal stresses such as low pH, bile salts, and high osmolarity, but also influenced susceptibility to antibiotics and disinfectants [86]. This research has the potential to contribute to the development of new strategies to combat antibiotic resistance in *K. pneumoniae* and inspire the audience about the promising future of this field. Furthermore, deletion of the *phoB* gene decreased the strain’s susceptibility to ceftazidime, cefepime, ceftriaxone, ertapenem, carbenicillin, and quinolones [86]. In addition, in *E. coli*, the inner core of LPS, crucial for antibiotic resistance, undergoes non-stoichiometric modifications regulated by transcription factors like the RpoE sigma factor and the PhoB/R TCSs, including the incorporation of a P-EtN residue on the second Kdo by EptB transferase and glucuronic acid by the PhoB/R-inducible WaaH glycosyltransferase. In *E. coli* K-12, the *waaH* and *eptC* genes, which mediate these modifications, are positively regulated by the PhoB/R TCS. Growth conditions can induce significant shifts in LPS composition and the prevalence of different glycoforms [201].

The CusSR TCS, consisting of the sensor histidine kinase CusS and the response regulator CusR, has shown increased expression in tigecycline- and carbapenem-resistant *K. pneumoniae* [202]. The TCS, CusR, and CusS sense periplasmic copper ions and activate the CusR regulon, which regulates the expression of the *cusRS* and *cusCFBA* operons in *E. coli*. Mutations in the region D genes (*copB*, *cueO*, *cusR*, *cusS*, *copC*, *copD*) significantly reduce the survival of both biofilm and planktonic cells, suggesting that any inherent copper resistance of a biofilm would not hinder future copper resistance therapeutics. Several additional copper resistance proteins, including PcoB, CopB, CopD, CopC, CueO, PcoA, CusR, and CusS, have been identified as potential therapeutic targets for dealing with *A. baumannii* infections and other MDR bacterial pathogens [203]. This discovery inspires hope for developing new treatments in the fight against antibiotic resistance. In addition, when exposed to increasing concentrations of silver nitrate, *K. pneumoniae* strains with the *sil* operon mutate SilS, leading to overexpression of the SilCBA efflux pump, while strains without the *sil* operon adapt through mutations in CusS, with secondary mutations disrupting the OM porin OmpC [204].

The CrrAB TCS, associated with colistin resistance in *K. pneumoniae* [205], plays a crucial role in this specific aspect of antibiotic resistance. Genetic alterations in the *crrAB* genes are responsible for the upregulation of *pmrAB* via CrrC, resulting in the modification of lipid A and colistin resistance [206]. The mutation of CrrB in CrrAB could also mediate resistance to tigecycline and tetracycline by regulating an RND-type efflux pump gene (H239_3064; KexD protein) [207]. The overexpression of *kexD* is the resistance mechanism for odilorhabdin [208]. These genes are located next to each other, forming the *crrBAC-kexD* cluster, which might act as a colistin resistance-regulating gene cluster [205]. The *crrBAC-kexD* cluster is proposed to have been acquired by the ancestor of the *K. pneumoniae* complex from other bacterial species, and the cluster may have been repeatedly lost and re-acquired in *K. pneumoniae* strains according to a phylogenetic analysis by Kim et al. [209].

Moreover, *crrB* mutations result in chemical alterations in lipid A, enhancing both virulence and resistance, albeit at a fitness cost. Interestingly, these mutations also modify carbon metabolism, activating the pentose phosphate pathway [210]. In addition to these changes, the CrrB mutation influences the bacterial OM by adjusting the LPS and regulating the synthesis of acyl-glycerophosphoglycerols (acyl-PGs). The lipid A palmitoyltransferase (PagP) gene, which exhibits increased expression in the *crrB* mutant, transfers a palmitate chain from glycerophospholipids to PGs, producing acyl-PGs. This process enhances the hydrophobicity of the OM, bolstering resistance against specific CAPs [69]. This intricate interplay of genetic alterations underscores the need for rational use of polymyxins to prevent the emergence of highly resistant and virulent bacterial strains. These observations suggest that colistin resistance may involve broader metabolic and envelope remodeling, with potential implications for virulence and fitness. Clinically, they support stewardship and mechanism-informed therapy to reduce selection for highly resistant variants [210].

Overexpression of KvhA (KvhSA response regulator) has been associated with increased fosfomycin susceptibility but reduced susceptibility to selected beta-lactams, illustrating regulatory trade-offs across drug classes [211]. In *K. pneumoniae*, KvgSA/KvhSA and other regulators (e.g., RcsAB, CrrAB) integrate stress cues (including oxidative and iron-limiting conditions) with CPS production, virulence traits, and antibiotic susceptibility modulation [212]. To connect these *Klebsiella*-centered regulatory examples with broader Enterobacterales stress circuitry, we briefly use the EvgS/EvgA network as a mechanistic illustration of how cross-talk can coordinate efflux, envelope homeostasis, and drug-response trajectories.

The EvgS/EvgA system in *E. coli* provides a valuable model for understanding how regulatory cross-talk can integrate intrinsic and adaptive resistance layers in Enterobacterales. EvgS/EvgA coordinates stress adaptation with reduced intracellular antibiotic accumulation by inducing multiple efflux-related genes, including *emrKY*, *mdtEF* (formerly *yhiUV*), *tolC*, *acrAB*, and *mdfA*, either directly or indirectly through additional regulators and two-component systems [213]. Many of the corresponding exporters function via the outer-membrane channel TolC, linking regulatory activation to increased efflux capacity during stress. Moreover, EvgS/EvgA is reported to be homologous to the KvgS/KvgA system in *K. pneumoniae*, supporting the concept that related regulatory logic may contribute to stress survival and envelope homeostasis in *Klebsiella* under clinically relevant pressures (e.g., oxidative stress and iron limitation) [212,213]. Connector proteins further exemplify how two-component networks can be integrated: in *E. coli*, SafA links EvgS/EvgA with PhoQ/PhoP, modulating downstream stress-response programs and potentially influencing susceptibility to last-line agents in specific genetic backgrounds, including polymyxins/colistin [213].

Genome-wide analyses further indicate that EvgA directly controls discrete chromosomal clusters enriched in EvgA-binding motifs, including loci encompassing *mdtEF* and regions near *emrKY*, reinforcing the view that EvgA acts as a central hub connecting environmental sensing with multidrug export capacity [214]. Expression of these clusters is additionally shaped by global regulators, including the histone-like nucleoid structuring protein (H-NS) and the alternative sigma factor RpoS, highlighting multi-layered control of efflux-related physiology during stress [214]. In addition, the expression of the acrAB and emrAB multidrug exporter systems can be induced by diverse treatments, further emphasizing how stress-regulatory networks can couple efflux with broader envelope-associated adaptation [214]. Consistent with this interplay between regulatory adaptation, efflux, and envelope homeostasis, polymyxin pressure can select for coordinated remodeling of outer-membrane biogenesis and LPS trafficking pathways, as discussed below.

In the context of colistin resistance, the observed upregulation of the AcrAB-associated component AcrA and envelope biogenesis factors, including the Bam complex and LPS-assembly/outer-membrane components such as LptE, in colistin-serially passaged and *mcr-1* cells suggests coordinated remodeling of outer-membrane homeostasis under polymyxin pressure [215]. Because LPS trafficking is essential for maintaining the permeability barrier of Gram-negative bacteria, disruption of the Lpt pathway (including the periplasmic LptA-mediated transfer step) can compromise outer-membrane integrity and increase susceptibility to antimicrobials [116]. Collectively, these observations highlight the Lpt system as a mechanistically plausible envelope-targeting vulnerability and a potential direction for future anti-Gram-negative drug development [216].

More broadly, envelope remodeling under polymyxin pressure intersects with global stress-response pathways that shape mutation supply, persistence, and the pace of resistance evolution in Enterobacterales, including the SOS response.

While the SOS response—regulated by RecA and LexA—is traditionally viewed as a key driver of antibiotic resistance through DNA repair and stress-induced mutagenesis (Figure 2), recent findings challenge the assumption that disabling RecA can prevent resistance evolution. In *E. coli ΔrecA* mutants, the loss of RecA not only impairs canonical SOS signaling but also leads to upregulation of the global repressor H-NS. This suppresses antioxidant-related genes, resulting in excessive ROS accumulation and elevated mutational burden. Upon exposure to beta-lactam antibiotics, this mutagenic environment facilitates the rapid selection of resistant subpopulations carrying mutations in genes such as *acrB* or beta-lactamase *ampC*. Thus, resistance emerges through a two-step process: increased mutation supply due to oxidative stress and selective enrichment under antibiotic pressure [217]. This mechanism operates independently of classical SOS-driven mutagenesis and highlights a paradox. While RecA is central to DNA repair and SOS activation, its absence can accelerate resistance evolution via alternative stress pathways. Moreover, the SOS system still plays a critical role in bacterial survival, allowing a fraction of cells to persist under lethal conditions and repopulate when stress subsides [218]. Together, these insights underscore the complexity of bacterial stress responses and their contribution to multidrug resistance.

Collectively, these examples reinforce the central message of this review: diverse molecular routes ultimately converge on the same outcome—reduced adequate drug exposure and enhanced survival under antimicrobial pressure—explaining why many paths can lead to clinically relevant resistance.

## 3. Conclusions

Antibiotic resistance in *Klebsiella*, a major hospital-acquired pathogen, represents a complex and urgent global health challenge. Resistance arises from a dynamic interplay of intrinsic, acquired, and adaptive strategies. Intrinsic barriers, such as porin remodeling, efflux pump activity, and biofilm-associated lifestyles, provide a foundational defense, while horizontal gene transfer and mutational events continually expand the resistance repertoire. Adaptive stress responses further enhance survival under antimicrobial pressure, bridging the gap until stable resistance traits emerge. Although laboratory susceptibility testing and genome sequencing have identified many resistance determinants, they do not fully explain all instances of treatment failure, underscoring the need to better characterize additional phenomena, such as tolerance and persistence.

Notably, the clinical impact of these mechanisms is amplified by the global expansion of high-risk epidemic lineages that efficiently acquire and maintain resistance determinants, thereby driving healthcare-associated transmission. In *K. pneumoniae*, major high-risk clonal groups and sequence types—most prominently CG258 (including ST258/ST512, and the related ST11), as well as widely disseminated MDR lineages such as ST147, ST307, ST15, ST101, ST395, ST231, ST383, and emerging ST268—serve as recurrent backbones for ESBL and carbapenemase dissemination. In parallel, classical hypervirulent lineages (notably ST23 and ST65/ST86) remain clinically significant, and the convergence of antimicrobial resistance and hypervirulence represents a growing public-health concern.

Beyond the optimized use of current agents, adjunct concepts that target non-enzymatic mechanisms—most notably efflux—remain of interest but have not yet been translated into routine clinical therapy. The emergence of resistance to newer agents such as cefiderocol and ceftazidime/avibactam underscores the necessity of proactive stewardship and continuous resistance monitoring. Promising directions include integrating genomics with transcriptomics and machine-learning approaches to uncover cryptic resistance mechanisms and better predict evolutionary trajectories. Ultimately, a multi-pronged approach linking molecular research with translational and clinical applications is essential to slow the spread of drug-resistant *Klebsiella* and improve treatment outcomes.

## Figures and Tables

**Figure 1 antibiotics-15-00037-f001:**
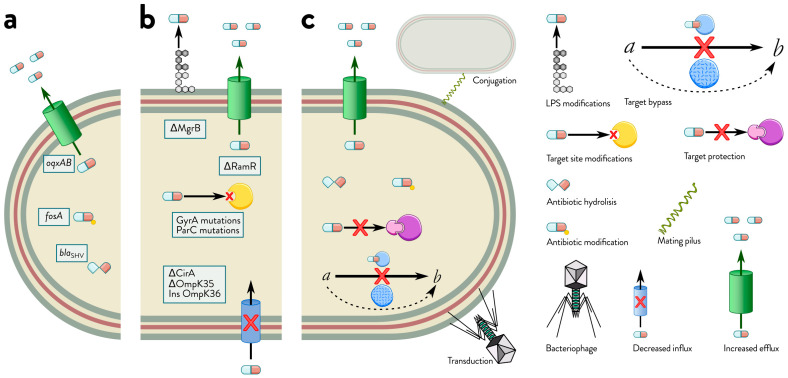
Overview of antibiotic resistance mechanisms in *K. pneumoniae*: efflux, enzymatic degradation, target alteration and horizontal transfer. This schematic illustrates major antibiotic resistance mechanisms in *K. pneumoniae*, integrating intrinsic, acquired, and adaptive strategies. Cell (**a**) demonstrates active efflux via the OqxAB pump and enzymatic inactivation through SHV-1 beta-lactamase and FosA glutathione transferase. Cell (**b**) shows porin loss (OmpK35/36), target-site mutations (GyrA, ParC), and gene deletions (MgrB, RamR) altering membrane permeability and antibiotic binding. Cell (**c**) highlights horizontal gene transfer via plasmid-mediated conjugation and phage-driven transduction facilitating resistance dissemination. Additional panels summarize critical resistance pathways, including LPS modification, antibiotic hydrolysis, and efflux pump upregulation, reflecting the multifactorial evolutionary adaptations of *K. pneumoniae* to antimicrobial pressure. The figure was created using the open-source software Inkscape v1.4.1 (https://inkscape.org/).

**Figure 2 antibiotics-15-00037-f002:**
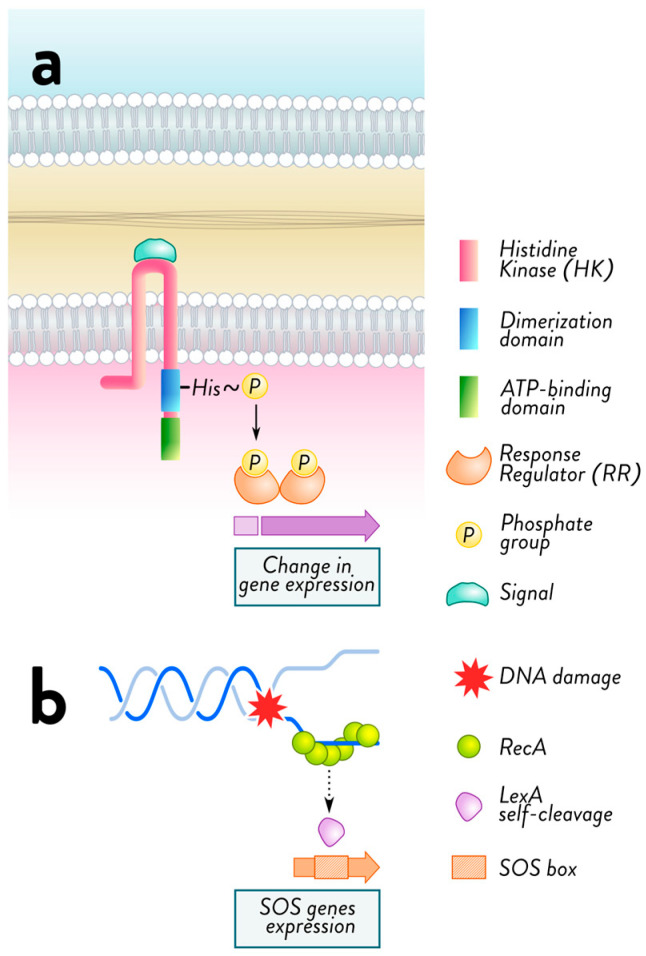
Bacterial regulation: two-component system and SOS response. The figure illustrates two key bacterial regulatory mechanisms. Panel (**a**) depicts a two-component signal transduction system in which histidine kinase (HK) detects environmental signals and phosphorylates a response regulator (RR), leading to changes in gene expression. Panel (**b**) represents the SOS response triggered by DNA damage, where RecA facilitates LexA self-cleavage, inducing the expression of SOS genes to repair the damage. The figure was created using the open-source software Inkscape v1.4.1 (https://inkscape.org/).

**Table 2 antibiotics-15-00037-t002:** Acquired antimicrobial resistance genes detected in *K. pneumoniae*.

Target Drug Class	Antimicrobial Resistance Genes
Aminoglycoside	*aac(3)-IIa*, *aac(3)-Ia*, *aac(3)-IId, aac(3)-IV*, *aac(6′)-33*, *aac(6′)-Iaf*, *aac(6′)-lb*, *aac(6′)-lb3*, *aac(6′)-lb4*, *aac(6′)-lb7*, *aac(6′)-lb-cr*, *aac(6′)-II*, *aac(6′)-IIa*, *aac(6′)-Iq*, *aadA*, *aadA1*, *aadA2*, *aadA5*, *aadA11*, *aadA16*, *aadA22*, *ant(2″)-Ia*, *aph(3′)-IIa, aph(3′)-VI*, *aph(3′)-VIb*, *armA*, *rmtB*, *rmtC*, *strA*, *strB*
Beta-lactam	ESBLs: ***bla*_BEL-1_**, *bla*_CTX-M-1_, *bla*_CTX-M-2_, *bla*_CTX-M-3_, *bla*_CTX-M-14_, *bla*_CTX-M-15_, *bla*_CTX-M-24_, *bla*_CTX-M-27_, *bla*_CTX-M-55_, *bla*_CTX-M-62_, *bla*_CTX-M-63_, *bla*_CTX-M-65_, *bla*_CTX-M-71_, *bla*_CTX-M-104_, *bla*_CTX-M-125_, ***bla*_KLUC-5_**, *bla*_KPC-12_, *bla*_KPC-14_, *bla*_KPC-25_, *bla*_KPC-33_, *bla*_OXA-2_, *bla*_PER-1_, *bla*_PER-7_, *bla*_SFO-1_, *bla*_SHV-2a_, *bla*_SHV-5_, *bla*_SHV-7_, *bla*_SHV-12_, *bla*_SHV-30_, *bla*_VEB-1_, *bla*_VEB-3_ Carbapenemases: ***bla*_BKC-1_**, ***bla*_GES-5_**, ***bla*_GIM_(?)**, *bla*_IMP-1_, *bla*_IMP-4_, *bla*_IMP-20_, *bla*_IMP-38_, *bla*_IMP-68_, *bla*_KPC-1_, *bla*_KPC-2_, *bla*_KPC-3_, *bla*_KPC-4_, *bla*_KPC-41_, *bla*_NDM-1_, *bla*_NDM-3_, *bla*_NDM-4_, *bla*_NDM-5_, *bla*_NDM-6_, *bla*_NDM-7_, *bla*_NDM-19_, *bla*_OXA-48_, *bla*_OXA-181_, *bla*_OXA-204_, *bla*_OXA-232_, *bla*_OXA-244_, ***bla*_SIM-1_**, *bla*_VIM-1_, *bla*_VIM-4_, *bla*_VIM-27_ AmpC: *bla*_CMY-2_, *bla*_CMY-4_, *bla*_CMY-6_, *bla*_CMY-16_, *bla*_CMY-33_, *bla*_DHA-1_, *bla*_FOX-5_, ***bla*_MOX-1_**, ***bla*_MOX-2_** Other beta-lactamases: ***bla*_CARB-2_**, *bla*_OXA-1_, *bla*_OXA-9_, *bla*_OXA-10_, *bla*_OXA-21_, *bla*_FONA-5_, *bla*_LAP-2_, *bla*_SCO-1_, *bla*_TEM-1_, *bla*_TEM-30_, *bla*_TEM-122_, *bla*_TEM-210_
Colistin	*mcr-1*, *mcr-2*, *mcr-3*, *mcr-8*
Fluoroquinolone	*qepA2*, *qnrA1, qnrA3*, *qnrA6*, *qnrB1*, *qnrB2*, *qnrB4*, *qnrB6*, *qnrB9*, *qnrB17*, *qnrB19*, *qnrS1, qnrE2*,
Fosfomycin	*fosA3*, *fosA7*
MLS	*ereA*, *ereA2*, *erm*(42), *ermB*, *ermT*, *lnuF*, *lnuG*, *mef*(B), *mphA*, *mphE*, *msrE*, ***estT***
Phenicol	*catA1*, *catB2*, *catB3*, *catB4*, *catB11*, *catII*, *cmlA1*, *cmlA4, cmlA5*, *cmx*, *floR*
Rifamycin	*arr-2*, *arr-3*
Sulfonamide	*sul1*, *sul2*, *sul3*
Tetracycline	*tet(A)*, *tet(B)*, *tet(C)*, *tet(D)*, *tet(G)*, *tetR*
Tigecycline	*tet*(X4), *tmexCD1-toprJ1*
Trimethoprim	*dfrA1*, *dfrA5*, *dfrA7*, *dfrA8*, *dfrA12*, *dfrA14*, *dfrA15*, *dfrA16*, *dfrA17*, *dfrA23*, *dfrA25*, *dfrA27*, *dfrA30*, *dfrA35*

Legend: MLS—macrolide-lincosamide-streptogramin B; ESBLs—extended-spectrum beta-lactamases. Bold-type genes—in silico analysis. Notes: GenPeptID—BKC-1 (AKD43328), BEL-1 (AHD24679), GES-5 (ABI63577.1), GIM (WKK67628), MOX-1 (BAA02563), MOX-2 (CAB82578). GenBankID—CARB-2 (MH476540—70599…71513), KLUC-5 (MH476540—44240…45115), SIM (MH681289—82414…83154), EstT (NZ_ON390817—52519…53364). Question mark—unclear information about the localization of the gene (chromosome or plasmid).

## Data Availability

All information summarized in this review is derived from publicly available sources, including NCBI GenBank and RefSeq databases. No new experimental data were generated. Details of documented antimicrobial resistance genes in *K. pneumoniae* are presented in Table 2 as an illustrative synthesis of existing knowledge.

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
