# Peer review of "Antibiotic Resistance in *Klebsiella pneumoniae* and Related Enterobacterales: Molecular Mechanisms, Mobile Elements, and Therapeutic Challenges"

_antibiotics, 2026, doi:10.3390/antibiotics15010037_

Round 1
Reviewer 1 Report
Comments and Suggestions for Authors
This literature review manuscript gives an overview about antibiotic resistance in Klebsiella spp. The topic of this manuscript is an interesting issue, however, some parts in the text should be revised to get to a clear and comprehensive view on this topic.
Comments
- In this manuscript the beta-lactamases are described in many parts of the text and class B, class D are also mentioned. However, the Ambler classification of beta-alctamses are not mentioned, and it would be important to mention that class A, class B, class C, and class D all belong to the Ambler classification of beta-lactamases. Therefore, I suggest to authors to add a short, informative description of Ambler classification of beta-lactamases in the introduction part of this manuscript, with beta-lactamase examples of each class.
- High-risk clones are also mentioned in the abstract, but these should be described in the manuscript. High-risk clones are globally recognized, major carriers of different resistance genes, therefore, the major high-risk clones of K. pneumoniae as well as the most frequently detected sequence types (STs) should be presented in the text.
- Table 2 is not clear. Each "Target drug class" and corresponding resistance genes group should be separated properly (e.g with line) to avoid a confusion in this table
- What do you mean by that: "high (p)ppGpp levels"? Gpp does not appear in the abbreviation list, and it is not defined in the text. Please, clarify!
Author Response
Response to Reviewer #1
We thank the Reviewer for the constructive and insightful comments. We have revised the manuscript accordingly to improve clarity, completeness, and readability. Below, we respond point-by-point. Reviewer comments are reproduced verbatim, followed by our responses. All newly added/clarified text has been incorporated into the revised manuscript and is highlighted in yellow.
Comment 1
“In this manuscript the beta-lactamases are described in many parts of the text and class B, class D are also mentioned. However, the Ambler classification of beta-lactamases are not mentioned, and it would be important to mention that class A, class B, class C, and class D all belong to the Ambler classification of beta-lactamases. Therefore, I suggest to authors to add a short, informative description of Ambler classification of beta-lactamases in the introduction part of this manuscript, with beta-lactamase examples of each class.”
Response:
We agree and thank the Reviewer for this important suggestion. We have added a concise, informative description of the Ambler classification to the Introduction, explicitly stating that classes A–D correspond to the Ambler scheme, and clarifying the mechanistic distinction between serine beta-lactamases (classes A/C/D) and metallo-beta-lactamases (class B). We also added representative Klebsiella-relevant examples for each class (e.g., class A: TEM/SHV/CTX-M and KPC; class B: NDM/VIM/IMP; class C: AmpC families; class D: OXA-48-like and OXA-1-like). This addition strengthens the conceptual framework for interpreting beta-lactamase diversity and inhibitor coverage in the remainder of the review.
Where reflected: Introduction (new paragraph/subsection on “Ambler classification of beta-lactamases”).
Comment 2
“High-risk clones are also mentioned in the abstract, but these should be described in the manuscript. High-risk clones are globally recognized, major carriers of different resistance genes, therefore, the major high-risk clones of K. pneumoniae as well as the most frequently detected sequence types (STs) should be presented in the text.”
Response:
We fully agree and have expanded the Introduction with a dedicated paragraph on high-risk clones and their epidemiological relevance. We defined the term, explained their role in AMR gene dissemination, and listed major STs: CG258 (ST258/ST512, ST11) and other globally prevalent MDR lineages (ST147, ST307, ST15, ST101, ST395, ST231, ST383), plus emerging ST268. Classical hypervirulent STs (ST23, ST65, ST86) and AMR–hypervirulence convergence are also highlighted. Relevant references were added.
Where reflected: Introduction (new paragraph “High-risk clones and dominant STs of K. pneumoniae.
Comment 3
“Table 2 is not clear. Each ‘Target drug class’ and corresponding resistance genes group should be separated properly (e.g with line) to avoid a confusion in this table.”
Response:
We thank the Reviewer for highlighting this issue. We have reformatted Table 2 to improve readability and prevent ambiguity. Each target drug class is now clearly separated, and individual rows are visually distinguished using an alternating color scheme (every second row shaded in gray for clarity). In the β-lactam section, we introduced structured subsections: ESBLs, plasmid-mediated AmpC, carbapenemases, and other β-lactamases.
Where reflected: Revised Table 2.
Comment 4
“What do you mean by that: ‘high (p)ppGpp levels’? Gpp does not appear in the abbreviation list, and it is not defined in the text. Please, clarify!”
Response:
We agree and apologize for the lack of definition. We have revised the sentence to explicitly define (p)ppGpp as the stringent-response alarmone (guanosine tetra- and pentaphosphate) and clarified its role as part of stress-response physiology that can contribute to reduced susceptibility/tolerance phenotypes. We have also added (p)ppGpp to the Abbreviations list to ensure consistency.
Where reflected: Section on adaptive resistance/stress responses (sentence revised) and Abbreviations list (added entry for “(p)ppGpp” and others).
Reviewer 2 Report
Comments and Suggestions for Authors
Some tiny but substantial adjustments to the style are suggested. The title of the manuscript is too restricted right now and doesn't indicate how broad the review is. The title should be altered to properly reflect the breadth of organisms and mechanisms described, given the publication talks about more than simply Klebsiella species. Additionally, it discusses additional members of the Enterobacteriaceae family, including Salmonella and Yersinia. Also, make sure that acronyms and abbreviations (such as TCSs, OMPs, and MBLs) are properly explained the first time they are used and used in the same manner throughout the text. Some acronyms don't have any explanation when they are first used. Lastly, bacterial taxonomic names like Klebsiella spp., Enterobacteriaceae, Salmonella, and Yersinia should only be italicized when the complete species name is supplied, such as "Klebsiella pneumoniae." Use regular (non-italic) font for names of families or genera that are not specific. It is recommended to examine the formatting in a systematic approach throughout the article. These are small suggestions for changes that are aimed to improve the presentation and consistency of the work, not to modify the scientific content.
Author Response
Response to Reviewer #2
We thank the Reviewer for the careful reading of our manuscript and for the helpful suggestions aimed at improving presentation, style, and consistency. We have implemented all the requested changes across the revised version. All newly added/clarified text and formatting corrections are highlighted in yellow in the manuscript.
Comment 1
“Some tiny but substantial adjustments to the style are suggested. The title of the manuscript is too restricted right now and doesn't indicate how broad the review is… it discusses additional members of the Enterobacteriaceae family, including Salmonella and Yersinia.”
Response:
We agree. We have revised the manuscript title to reflect better the broader scope of organisms and mechanisms addressed, beyond Klebsiella. In addition, we added a concise scope statement in the Introduction clarifying that, while the review focuses primarily on Klebsiella, selected examples from other clinically relevant Enterobacterales (including Salmonella and Yersinia) are discussed when they illustrate shared mechanisms, mobile genetic platforms, or clinically relevant therapeutic implications.
Where reflected: Title page and Introduction (new scope sentence/paragraph).
Comment 2
“Also, make sure that acronyms and abbreviations (such as TCSs, OMPs, and MBLs) are properly explained the first time they are used and used in the same manner throughout the text. Some acronyms don't have any explanation when they are first used.”
Response:
We agree and have addressed this systematically. We reviewed the manuscript for abbreviations and ensured that all acronyms are defined at first mention and subsequently used consistently throughout the text. In particular, we standardized key recurring abbreviations by introducing them in the singular form (e.g., two-component system (TCS), outer membrane protein (OMP), metallo-beta-lactamase (MBL)) and then using singular/plural forms consistently (TCS/TCSs, OMP/OMPs, MBL/MBLs) according to grammatical context. We also verified consistency between the main text and the Abbreviations list.
Where reflected: Throughout the manuscript (first mention of each acronym corrected) and Abbreviations list (expanded/updated where needed).
Comment 3
“Lastly, bacterial taxonomic names like Klebsiella spp., Enterobacteriaceae, Salmonella, and Yersinia should only be italicized when the complete species name is supplied… Use regular (non-italic) font for names of families or genera that are not specific. It is recommended to examine the formatting in a systematic approach throughout the article.”
Response:
We appreciate the Reviewer’s emphasis on taxonomic consistency, and we have implemented a systematic, manuscript-wide review of formatting. To ensure alignment with the journal’s style guidance (MDPI layout instructions), we standardized nomenclature such that genus and species names are italicized (e.g., Klebsiella pneumoniae), abbreviated genus names are used after first mention (e.g., K. pneumoniae), and “spp.” is kept in roman type (e.g., Klebsiella spp.). We also standardized higher taxonomic ranks (e.g., Enterobacteriaceae/Enterobacterales) consistently in Roman type throughout the manuscript.
Where reflected: Throughout the manuscript (systematic revision of taxonomic typography).
Reviewer 3 Report
Comments and Suggestions for Authors
Dear Authors, The review manuscript titled “Escalating antibiotic resistance in Klebsiella spp. and beyond: from molecular strategies to therapeutic challenges” was authored by Zdarska et al. for publication with the Journal “Antibiotics”. Authors compiled the review on the critical issue of highly drug-resistant superbugs of the Klebsiella pneumoniae species posing a severe public health crisis by exhibiting a complex, multi-layered defense system. In addition, they touch on the intrinsic resistance (like antibiotic-expelling efflux pumps and porin loss), the acquisition of destructive genes through horizontal transfer (such as those encoding carbapenemases and beta-lactamases), and the formation of protective biofilms on medical equipment. They highlight that the defenses work together synergistically to ensure bacterial survival. Overcoming this crisis requires continued research into the resistance landscape to guide the development of new treatments, including efflux pump inhibitors and novel antibiotic/inhibitor combinations, to curb the global spread of these dangerous bacteria. The review manuscript provides an intriguing and insightful analysis of the molecular and genomic aspects, as well as the therapeutic measures and challenges encountered during this process. The challenges are thoroughly addressed, and the relationship between the plasmid, genes, and the resulting product after expression is clearly outlined. The manuscript could be enhanced by implementing the suggestions below: Abstract: The abstract lacks details on molecular mechanisms, as indicated in the title. The authors should consider incorporating this information. Line 126: Open bracket before “specifically” Line 133 to 134: The authors should concentrate on Klebsiella spp. while mentioning other bacterial species; however, the subtitle should specifically reference Klebsiella Plagiarism on first 2 paragraphs of section 2 in Line 135 to Line 157. Line 290 to 298 and 354 to 372: The focus is on Pseudomonas aeruginosa and Staphylococcus aureus, respectively, and the authors should aim to stay centered on this organism of interest. The subsections appear quite general; making them more specific could enhance readers' understanding, ensuring that important information is clear and not buried. The content is readable, but the flow feels somewhat inconsistent, particularly between lines 851 and 862, which seem out of place. Mention the sources of figures on their figure titles.
Author Response
Response to Reviewer #3
We thank the Reviewer for the constructive and insightful comments. We have revised the manuscript accordingly to improve specificity, focus on Klebsiella pneumoniae, and overall readability/flow. Below, we respond point-by-point. Reviewer comments are reproduced verbatim, followed by our responses. All newly added or clarified text has been incorporated into the revised manuscript and is highlighted in yellow.
Comment 1
“Abstract: The abstract lacks details on molecular mechanisms, as indicated in the title. The authors should consider incorporating this information.”
Response:
We agree. We revised the Abstract to explicitly include key molecular mechanisms relevant to K. pneumoniae, including reduced permeability via OmpK35/OmpK36 remodeling, multidrug efflux (AcrAB-TolC/OqxAB), enzymatic drug inactivation (ESBLs and carbapenemases), and clinically meaningful pathways of polymyxin/colistin resistance (e.g., mgrB inactivation and PhoPQ/PmrAB-mediated lipid A modification). We also briefly mention adaptive tolerance/persistence and biofilm-associated survival to align with the manuscript’s scope and title.
Where reflected: Abstract (revised).
Comment 2
“Line 126: Open bracket before ‘specifically’ ”
Response:
We thank the Reviewer for noting this. We corrected the punctuation.
Where reflected: Introduction/early text (line 126 in the tracked version).
Comment 3
“Line 133 to 134: The authors should concentrate on Klebsiella spp. while mentioning other bacterial species; however, the subtitle should specifically reference Klebsiella”
Response:
We agree. We refined the scope statements and adjusted section/subsection wording so that Klebsiella pneumoniae (and Klebsiella spp. where appropriate) remains the primary focus throughout the manuscript. References to other Enterobacterales (e.g., Salmonella, Yersinia) were retained only where they illustrate conserved mechanisms or mobile genetic platforms directly relevant to Klebsiella, and the relevant subtitle(s) were revised to reflect this focus more explicitly.
Where reflected: Title/subtitles and Introduction scope statements (revised wording).
Comment 4
“Plagiarism on first 2 paragraphs of section 2 in Line 135 to Line 157.”
Response:
We appreciate this critical point. To address any potential similarity to general textbook phrasing, we rewrote the first two paragraphs of Section 2 in an entirely original manner. We re-framed them as a Klebsiella-specific “layered resistance” conceptual introduction. The revised text emphasizes how intrinsic barriers, acquired determinants, and adaptive programs converge in clinical K. pneumoniae isolates, providing a more transparent bridge into the subsequent mechanism-focused subsections.
Where reflected: Section 2 (opening paragraphs rewritten; lines 135–157 in the tracked version).
Comment 5
“Line 290 to 298 and 354 to 372: The focus is on Pseudomonas aeruginosa and Staphylococcus aureus, respectively, and the authors should aim to stay centered on this organism of interest.”
Response:
We agree. We removed organism-divergent examples centered on Pseudomonas aeruginosa (AmpC-related discussion) and Staphylococcus aureus (NorA efflux example), and replaced them with text focused on K. pneumoniae/Enterobacterales-focused text. The revised passages now emphasize the clinical relevance of porin remodeling (OmpK35/OmpK36), efflux systems (AcrAB-TolC/OqxAB), and their interaction with acquired β-lactamases in Klebsiella. The EPI discussion was retained but re-centered on Gram-negative/Enterobacterales considerations and limitations to ensure relevance.
Where reflected: Section 2.1/2.2 (revised passages replacing Pseudomonas and S. aureus examples).
Comment 6
“The subsections appear quite general; making them more specific could enhance readers' understanding, ensuring that important information is clear and not buried.”
Response:
We agree and addressed this by implementing targeted, high-impact edits without altering the scientific scope. Specifically, we (i) revised subsection titles and opening sentences to be more K. pneumoniae-specific, (ii) added short “anchor” sentences highlighting the most clinically relevant Klebsiella examples and synergies, and (iii) inserted brief “take-home” and linking (micro-bridge) sentences to guide the reader through the progression from intrinsic barriers → acquired determinants → adaptive programs → mechanistic interplay.
Where reflected: Throughout Sections 2.1–2.4 (subsection framing, anchor sentences, and micro-bridges added).